# Integrated Membrane–Electrocoagulation System for Removal of Celestine Blue Dyes in Wastewater

**DOI:** 10.3390/membranes10080184

**Published:** 2020-08-13

**Authors:** Muhammad Syaamil Saad, Lila Balasubramaniam, Mohd Dzul Hakim Wirzal, Nur Syakinah Abd Halim, Muhammad Roil Bilad, Nik Abdul Hadi Md Nordin, Zulfan Adi Putra, Fuad Nabil Ramli

**Affiliations:** 1Chemical Engineering Department, Universiti Teknologi PETRONAS, Seri Iskandar, Perak 32610, Malaysia; syaamil_19001012@utp.edu.my (M.S.S.); lila_19634@utp.edu.my (L.B.); nur_17005248@utp.edu.my (N.S.A.H.); mroil.bilad@utp.edu.my (M.R.B.); nahadi.sapiaa@utp.edu.my (N.A.H.M.N.); fuad.nabil_23160@utp.edu.my (F.N.R.); 2PETRONAS Group Technical Solutions, Process Simulation and Optimization, Level 16, Tower 3, Kuala Lumpur Convention Center, Kuala Lumpur 50088, Malaysia; zulfan.adiputra@petronas.com.my

**Keywords:** electrocoagulation, nanofiber membrane, celestine blue, integrated system

## Abstract

The textile industry provides for the needs of people especially in apparel and household items. The industry also discharges dye-containing wastewater that is typically challenging to treat. Despite the application of the biological and chemical treatments for the treatment of textile wastewater, these methods have their own drawbacks such as non-environment friendly, high cost and energy intensive. This research investigates the efficiency of the celestine blue dye removal from simulated textile wastewater by electrocoagulation (EC) method using iron (Fe) electrodes through an electrolytic cell, integrated with nylon 6,6 nanofiber (NF) membrane filtration for the separation of the flocculants from aqueous water. Based on the results, the integrated system achieves a high dye removal efficiency of 79.4%, by using 1000 ppm of sodium chloride as the electrolyte and 2 V of voltage at a constant pH of 7 and 10 ppm celestine blue dye solution, compared to the standalone EC method in which only 43.2% removal was achieved. Atomic absorption spectroscopy analysis was used to identify the traces of iron in the residual EC solution confirming the absence of iron. The EC-integrated membrane system thus shows superior performance compared to the conventional method whereby an additional 10–30% of dye was removed at 1 V and 2 V using similar energy consumptions.

## 1. Introduction

The textile industry in Malaysia is declared as 13th largest exporting industry in 2018 with exports products worth approximately MYR 12 billion [1]. It is not a new industry as Malaysia has been actively involved in textile manufacturing since 1980. With regards to its contribution towards the growing economy of the country, the textile industry also produces wastewater. This waste that contains a 20% amount of dyes that did not bind to the textile or cloth, which is discharged as effluent [2]. Dye is among the key components in textile manufacturing as it is used to paint clothes with aesthetic colours. However, the usage of dye and its presence in the effluent from the textile industry poses a threat to the ecological environment [3]. The textile industry has caused environmental pollution due to the usage of more than 2000 types of chemicals and over 10,000 dyes in the manufacturing process [4]. The colour intensity of the water significantly affects the public notion as the presence of an unnatural colour is aesthetically undesired. Most of the dyes used in the textile industries are known as azo-reactive dyes, which contain the azo/keto-hydrazone group that acts as a good solubilizing component [5]. Due to the relatively low level of the dye–fibre fixation property, 50% of the dyes tend to passage out together with the effluents into the aqueous environment. Furthermore, the high stability and xenobiotic nature of the azo dyes cannot be easily decomposed biologically, since they persist in the environment for a long time [6]. For example, hydrolyzed dye Reactive Blue 19 has a lifetime up to 46 years at a pH of 7 and at a temperature of 25 °C [7].

Currently, there are wide ranges of treatments used in the industry for removing dye in wastewater, such as biological treatment [8,9,10], fenton oxidation [11,12], membrane separation [13,14] and physiochemical treatment [15,16]. However, these methods pose a few major drawbacks, such as their high operation cost, larger formation of sludge, time consuming (long retention time) and the production of toxic by-products [17,18]. Another approach yet effective in treating wastewater containing dyes is the electrocoagulation (EC) method [19,20]. The EC method is an electrochemical process that applies direct current to the electrodes, normally iron or aluminium, dipped into the electrolytic solution containing the wastewater to be treated. The simultaneous formation of hydroxyl ions and hydrogen gas at the cathode generate the coagulant by the oxidation of the anode [21]. The gases (H_2_, O_2_) produced at the cathode and anode give a floatation effect, separating the pollutants to the floc-foam layer on the water surface [22]. The reactions that occur on the iron electrode were stated by [23] as in Equations (1)–(4):
Anode$Fe\left(s\right)\to 4\;Fe^{2+}\left(aq\right)+2\;e^{-}$(1)At alkaline condition$Fe^{2+}+2\;OH^{-}\to Fe{\left(OH\right)}^{2}$(2)At acidic condition$4\;Fe^{2+}+O_{2}+2\;H_{2}O\to 4\;Fe^{3+}+4\;OH^{-}$(3)Cathode$2\;H_{2}O+2e^{-}\to H_{2}+2\;OH^{-}$(4)

Kabdaşlı et al. stated that the dissolved metal cations from the anode combines with the hydroxyl ions to form metal hydroxide and eventually neutralizes the suspended particles in the water by forming the monomeric and polymeric hydroxo complex species [24]. The pollutants in the wastewater can be removed either through complexation or electrostatic attraction later by the remaining Fe(OH)n [25]. The generation of Fe(OH)n(s) can be in the form of divalent and trivalent that are highly dependent on the pH and the potential of the aqueous medium, thus the optimum pH of the water needs to be maintained [26].

In addition, membrane technology has been widely used for separation purposes in most industrial processes, especially for water treatment [27,28,29]. The technology is known to be simple, effective, requires low energy, is cost effective and no chemical additives or phase changes are required [30]. Various fields have been implementing these technologies at large scales, namely gas purification [31,32], food processing [33,34], wastewater treatment [35,36] and in the pharmaceutical industry [37,38]. The type of membrane is usually categorized based on the pore size and/or the driving force that causes the separation to take place [39,40]. Most of the membranes used in the industries are organic polymer and the nanofiber (NF) membrane is one of them [41].

Nanofibers have been widely explored for many applications such as energy storage, health care, biotechnology, information technology and more so due to their large surface to volume ratio property, light weight, hydrophilicity, good mechanical property and integral porous structure [42]. It was reported that membrane-based wastewater treatment is an excellent approach to handle the large volume of sludge formation in certain processes [43]. Thus, since the EC method for wastewater treatment dyes produces flocculants on the surface of the water at the end of the treatment, the membrane separation process can be opted to separate the flocculants completely from the wastewater, leaving purified water as the permeate. In the context of the treatment of textile wastewater via EC, the incorporation of the membrane-based process is expected to enhance the effluent quality. As a post treatment to EC, a nanofiber membrane can be used to filter the suspended floc formed during the EC process. It is hypothesized that the integration of nylon 6,6 nanofiber-based membrane filtration with EC for the treatment of textile effluents would result in high dye removal efficiency. A nylon 6,6 nanofiber-based membrane poses good mechanical strength, good hydrophilicity and is thus suitable for this application [44].

This study proposes an integrated EC and membrane filtration for dye removal from textile-based wastewater. More specifically, it is focused on the integrated system aimed for a low energy input (low voltage) to treat synthetic dye wastewater containing celestine blue. To the best of our knowledge, there is no report available on EC and the membrane filtration integration using nylon 6,6 nanofiber membrane. The objectives of this study are to (i) evaluate the treatment of the dye-contaminated wastewater using a standalone EC method with an iron (Fe) electrode, (ii) to evaluate the influence of the voltage and amount of electrolyte on the process efficiency on the standalone EC and lastly (iii) to integrate the EC with membrane filtration for enhancing the dye removal efficiency.

## 2. Materials and Methods

### 2.1. Chemicals and Reagents

The chemicals used were celestine blue (C_17_H_18_CIN_3_O_4_) (Sigma-Aldrich, St. Louis, MO, USA). Sodium chloride (NaCl) (Merck, Kenilworth, NJ, USA) was used as the electrolyte in the aqueous solution. Nitric acid (Merck, Kenilworth, NJ, USA) was used as the cleaning agent to remove oxidized iron on the electrode surface. Chemicals that were used for membrane preparation include formic acid (98–100%) (Merck, Kenilworth, NJ, USA), acetic acid (Merck, Kenilworth, NJ, USA) brand and nylon 6,6 pellets (Sigma-Aldrich, St. Louis, MO, USA).

### 2.2. Preparation of Stock Solutions

The stock solution was prepared as 100 ppm (mg/L) celestine clue dye solution, by weighing 100 mg/0.1 g of celestine blue dye and diluting it with 1000 mL of distilled water. The, 10 ppm dye solution was prepared from the stock solution for the wastewater treatment, and the dilution method was applied to obtain the volume of stock solution to be used.

### 2.3. Equipment and Apparatus

A direct current (DC) power supply was used to supply the DC potential to the synthetic wastewater to promote the coagulation process. Crocodile clips, retort stands and clamps were responsible to hold the electrodes in place while the magnetic stirrer was used to agitate the synthetic wastewater throughout the experiment. The filter paper was placed on a filter funnel to separate the sludge formed from the synthetic wastewater. The initial and final absorbance of the wastewater tested was obtained using an UV–Vis spectrophotometer. Figure 1 shows the basic experiment set up for the integrated membrane–EC process.

### 2.4. Instrument Analysis

The UV–Vis spectroscopy analysis was conducted to determine the percentage removal of dye by measuring the absorbance. The initial concentration of the synthetic wastewater was compared to the treated wastewater using an integrated EC–membrane system. Atomic absorption spectroscopy (AAS) was used to measure the concentration of the chemical element and detect any presence of metal such as iron in the synthetic wastewater after EC by using a flame atomic absorption method.

### 2.5. Nylon 6,6 Nanofiber Membrane Synthesis and Characterization

The polymer solution was prepared with a solvent to polymer ratio of 86:14. A basis of 20 mL of solution was used to calculate and measure the amount of solvent and polymer used. Acetic acid and formic acid were added in a 50:50 ratio along with the nylon 6,6 pellets to form the solution. The solution was prepared one day prior to electrospinning process to ensure the homogeneity of the liquid. The electrospinning method was adopted to a synthesis nylon 6,6 nanofiber membrane. Then, a 5 mL syringe filled with the polymer solution was mounted on a syringe pump and placed 15 cm from a spinning cylindrical collector while the tip of the needle was connected to a high voltage power supply. The solution was injected with a flowrate of 0.4 mL/h, 20 kV supplied and 500 rpm of collector rotation speed.

The morphology of the nanofiber membrane was characterized using a field emission scanning electron microscope (VPFESEM, ZeissSupra55 VP, ZEISS Sigma, Jena, Germany) and the average surface roughness using an atomic force microscopy (AFM, Model: NanoNavi E-Sweep, Bruker, Billerica, MA, USA). The pore size of the membrane was determined using ImageJ while the porosity of the resulting membranes was determined by using the dry wet method.

### 2.6. Experiment Procedure

The experiments were conducted in a batch electrolytic cell by using 800 mL of celestine blue dye as synthetic wastewater. The temperature (room condition), pH 7 and initial concentration (10 ppm) of the dye were maintained constant throughout all the experiments. Before starting the experiment, 10 ppm of celestine blue dye solution was prepared. Then, a 1 L beaker was used to contain 800 mL of the synthetic wastewater.

In the first part of the experiment, the synthetic wastewater was treated using only a nylon 6,6 nanofiber membrane. The trapezium shaped flow channel used had an effective surface area of 9 cm^2^ and was assembled between rubber and a polyvinyl chloride (PVC) frame by enveloping the edges using polytetrafluoroethylene (PTFE) tape and clipped with big paper clippers to avoid leakages. The assembly of the membrane filtration module can be seen in Figure 2.

The feed and retentate tube connecting the membrane were immersed into the beaker containing synthetic wastewater whereas the permeate tube was channelled into a 500 mL measuring cylinder. The pressure was set constant at the 0.1 bar by adjusting the frequency on the water pump and the experiment was carried out at the desired parameters. The dye removal efficiency was calculated from the feed and permeate absorbance obtained from spectrographic analysis. The flux (*J*) was calculated using Equation (5), where *V* is the volume (L), *A* is the effective membrane area (m^2^) and *t* is the time (h):(5)$$J=\frac{V}{At}$$

The volume of the permeate was collected at 2 min intervals and the experiment was conducted for 60 min at constant celestine blue dye concentration of 10 ppm, pH 7 and at room temperature. For every parameter involved, the membrane permeability was measured trice and the results are presented as the averages. The dye removal efficiency was calculated from the absorbance measured using Equation (6):Dye Removal = (C_0_ − C_1_)/C_0_ × 100%(6)
where C_0_ is the initial absorbance measured and C_1_ is the final absorbance measured. The experiment was then repeated but with EC incorporated with the membrane method as shown in Figure 1. Samples were taken every 2 min for the UV–Vis and AAS analyses.

### 2.7. Electrocoagulation Method

The optimization studies on the EC of dye in wastewater was done in previous research [23]. The parameters used to measure the removal efficiency of dyes was the voltage supplied to the EC with a value of 2, 4, 6, 8 and 10 V. Current densities reported by Balasubramaniam et al. at 1, 2 and 10 V are 5.71 mA/cm^2^, 6.29 mA/cm^2^ and 24.29 mA/cm^2^, respectively. The total effective electrode area was 105 cm^2^ with a 2 cm interelectrode gap. The concentrations of electrolyte (NaCl) were also varied with values of 250, 500, 750, 1000 and 1250 ppm. Based on the studies, 1000 ppm of NaCl and the voltage at 1 V was observed as the best condition for EC with a 43.2% removal of the dye and 0.75 kWh/m^3^ of energy consumed.

## 3. Results and Discussion

### 3.1. Characteristics of Nylon 6,6 NANOFIBER membrane

The morphology of the membrane and the surface roughness are shown in Figure 3a,b, respectively. The FESEM analysis has shown that the average diameter of the fibre strand of the synthesized membrane is 746.4 ± 50 nm. An average surface roughness of 56.5 ± 10 nm was obtained from the AFM analysis which is in agreement with the nanofiber membrane properties [45]. The pore size and porosity of the membrane are shown in Table 1.

### 3.2. Permeability of Nylon 6,6 Nanofiber Membrane

The nylon 6,6 nanofiber membrane was evaluated for the filtration of the synthetic wastewater to measure the membrane permeability and the dye rejection as shown in Figure 4. The results will be used as the reference for the integrated process. In Figure 4a, the flux of liquid passing through the membrane decreased exponentially and remained almost constant after 10 min. The fouling effect was observed to take place as the pollutant started to block the pores of the membrane which make the flux decrease and is common to find in nanofiber membrane [46]. This was also supported by the results from the percentage removal of synthetic wastewater in Figure 4b. The range of dye removal increases from 6.4% to 8.4%, which indicates that as fouling happens, the percentage of dye removal increases due to more dye particles being trapped in the membrane pores and hence lowers the membrane pore size. It is evident that most of the synthetic wastewater tends to entrain across the membrane due to the small size and the amount of permeate collected does not change significantly [47].

### 3.3. Integrated Membrane–Electrocoagulation System

In a previous study by Balasubramaniam et al. [23], the voltage and NaCl concentration were explored to determine the optimum operating condition of the EC of celestine blue. A few optimized conditions that were considered from the results obtained from the previous studies are 250, 500 and 1250 ppm of sodium chloride (NaCl) at 1 V and 250, 1000 and 1250 ppm at 2 V [23]. A lower voltage is considered the best due to the low energy consumption with satisfying the dye removal percentage. The results from our previous studies are summarized in Table 2.

The EC system was integrated with membrane filtration at a voltage of 1 and 2 V, pH 7, and 10 ppm of initial concentration of celestine blue dye. Based on the results in Figure 5, when the concentration of NaCl increases, the overall dye removal efficiency trend increases with time and the percentage removal is higher when compared to the results obtained without integrating the membrane within the EC system. With the addition of nanofiber-based microfiltration as a post treatment in the integrated system, more agglomerated dye particles are entrapped on the membrane surface. This allows clearer water to be produced in the permeate [16,48]. On another note, at 1 V, the highest removal efficiency of 72.0% is achieved at the 60th minute using 1250 ppm NaCl, followed by 68.1% using 500 ppm NaCl and the lowest removal efficiency of 54.2% using 250 ppm NaCl. Meanwhile, at 2 V, the highest dye removal efficiency of 79.4% was observed at the 60th minute, also using 1250 ppm NaCl, followed by 75.6% using 1000 ppm NaCl and the least dye removal efficiency of 60.8% was obtained using 250 ppm NaCl. It can be concluded that the removal efficiency is the highest when the highest concentration of NaCl is used which was 1250 ppm because the higher the amount of Cl^−^ ions present in the synthetic wastewater, the higher the dissolution rate in anode to produce the metal coagulant in the wastewater [49]. It is also reported that the addition of NaCl of more than 1250 ppm can further reduce the amount of iron concentration in wastewater through standalone EC [50]. The highly available anions (Cl^−^) can decrease the produced positive charge of iron ions that enlarge the flocs compared to the ones formed under low NaCl concentration [51]. Subsequently, a large floc eases the membrane filtration process due to a highly porous cake formation. It is also evident that the dye removal efficiency is higher at 2 V than at 1 V due to the formation of bigger flocs and more production of metal hydroxide ions for the EC reaction. Despite its efficacy in enhancing dye removal efficiency, it is worthwhile to mention that high NaCl concentration will also necessitate a further treatment process, since the direct discharge of saline water on the surface water body is not allowed.

Figure 6 shows the permeability obtained for integrated membrane EC system at 1 V and 2 V respectively. Based on the graphs at all parameters tested, it is evident that the flux of the membrane filtration when incorporated in the integrated systems reaches the steady state quicker when compared to the standalone filtration system. The flux drops gradually over time and reaches the steady state in the middle of the experiment until towards 60th minute. The highest permeate flux drop is observed using 1250 ppm of NaCl at both 1 V and 2 V due to rapid formation of flocs and the lowest permeate flux decline is observed using 250 ppm of NaCl at both 1 V and 2 V as well. All tests show the consistent declining trend. However, higher fluxes were achieved for standalone filtration system in both 1 V and 2 V condition. This clearly indicates that membrane fouling has taken place due to the membrane pore clogging and formation of cake layer by the floc formed in the EC on the membrane surface. This is supported by study on EC flocs from iron electrode using kaolin suspension where the diameter of flocs was reported to be 141 ± 4 µm [52]. As for iron flocs from dyes as wastewater, insufficient information was provided in regards of the floc size. As can be seen from the graph, it is noticed that the initial permeate flux was high because no formation of flocs in the beginning time of EC and the pores are clean and opened at 0th time [53]. However, the permeate flux started to drop significantly indicating flocs are being produced due to EC process and it is being subsequently filtered through the membrane, resulting quick clogging the membrane pores. Further decline in flux is due to the more formation of flocs over time during EC, creating additional resistance to the permeate flow.

The dye removal achieved by the integrated EC–membrane system is found to be among the highest reported to date, when compared to a few studies done on dye removal with different methods (Table 3). In EC, the removal of dye was observed to be 43.2%. In many of the reported studies, the EC were operated at a much higher voltage, thus optimization was done by [23] with voltage and NaCl concentration. Adsorption with carbon nanofibers were done by [54] using methylene blue as feed. The removal rate observed was 77.8%, still below the value obtained in this study. While integrated bio-electrochemical and anaerobic systems exhibit a 97.5% removal of dye, the configuration of the reactor is more complex and the results obtained are not instantaneous. The treatment requires weeks and up to a month to produce excellent results. With an integrated electrochemical and nanofiber membrane, a much simpler setup is achievable to produce 79.4% dye removal with low energy consumption.

## 4. Conclusions

This study assesses the dye removal trend from synthetic textile wastewater at low (i.e., 1 V and 2 V) and high voltages (i.e., 2 V, 4 V, 6 V, 8 V and 10 V) at different concentrations of NaCl using standalone EC, standalone membrane filtration and EC-integrated membrane systems. For the standalone EC method, 43.2% of the dye removal is achieved using 1000 ppm of sodium chloride at 1 V in 24 min with low energy consumption of 0.75 kWh/m^3^, without producing excess Fe ions in the effluent. For the standalone membrane filtration, the dye removal efficiency is very low and thus not feasible. Integrated EC–membrane filtration results in 79.4% of dye removal with only 2 V applied and corresponding to the energy input of 1.65 kWh/m^3^. The nanofiber membrane could be reused at least twice. Such a problem can be addressed by the development of an advanced nanofiber membrane with enhanced mechanical strength which will become the follow up study. Moreover, the EC-integrated membrane system improved the dye removal efficiency by 10–30% and the sludges formed during the EC also simultaneously separated. Hence, the treatment of dye wastewater using an integrated system can be concluded to be sufficiently effective, even at a lower voltage to outweigh the efficiency of the EC and membrane methods alone.

## Figures and Tables

**Figure 1 membranes-10-00184-f001:**
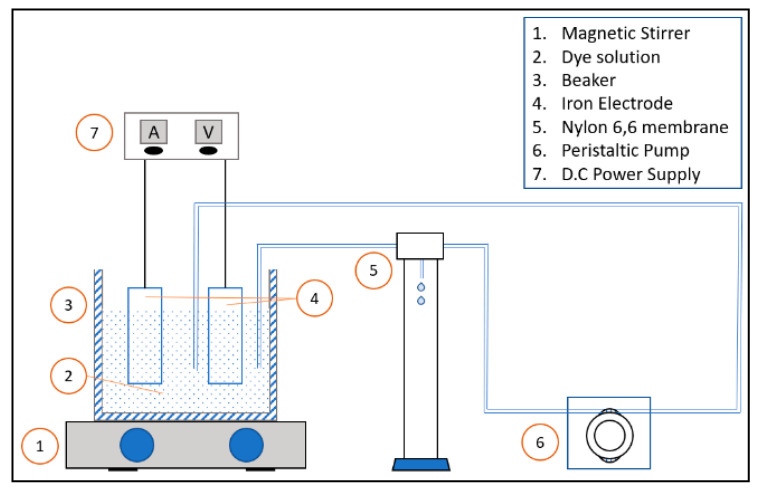
Schematic of the integrated system with the membrane as the post-treatment.

**Figure 2 membranes-10-00184-f002:**
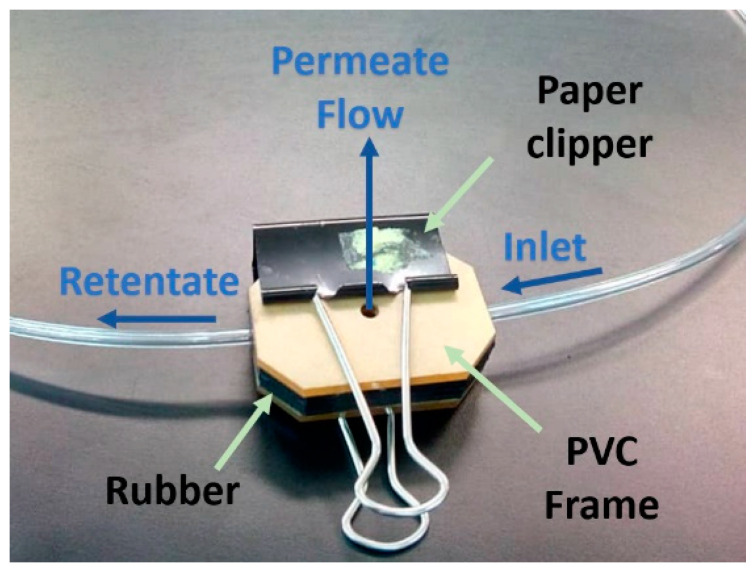
Membrane module with polyvinyl chloride (PVC) frame used for the microfiltration setup.

**Figure 3 membranes-10-00184-f003:**
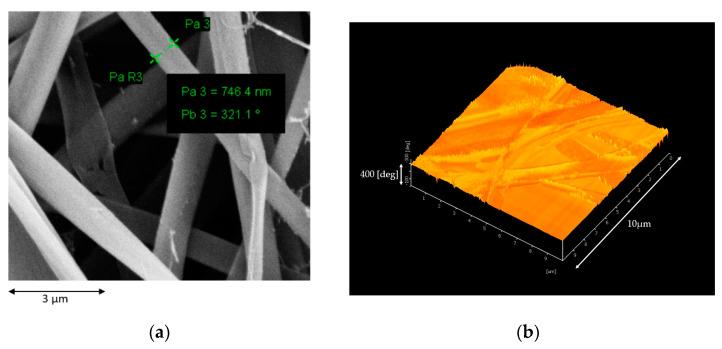
(**a**) Surface morphology of the membrane analysed using FESEM and (**b**) the surface roughness of the membrane by atomic force microscopy (AFM).

**Figure 4 membranes-10-00184-f004:**
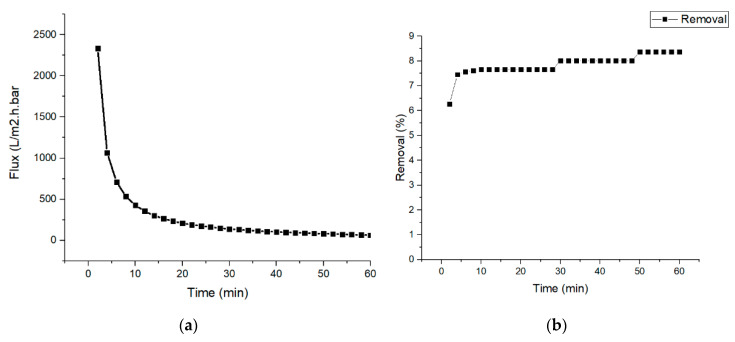
(**a**) Nylon 6,6 nanofiber membrane flux for the 10 ppm dye solution and (**b**) the percentage removal of the 10 ppm celestine blue dye by the nylon 6,6 nanofiber membrane.

**Figure 5 membranes-10-00184-f005:**
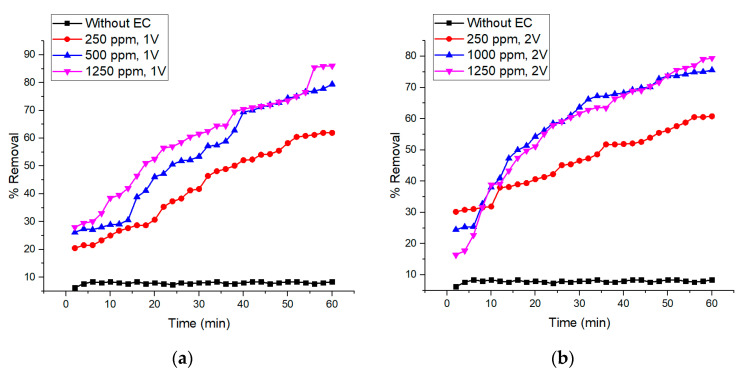
Comparison of the dye removal treatment with and without electrocoagulation (EC) using (**a**) 250, 500 and 1250 ppm of NaCl at 1 V and (**b**) 250, 1000 and 1250 ppm of NaCl at 2 V at pH 7 using 10 ppm of celestine blue solution.

**Figure 6 membranes-10-00184-f006:**
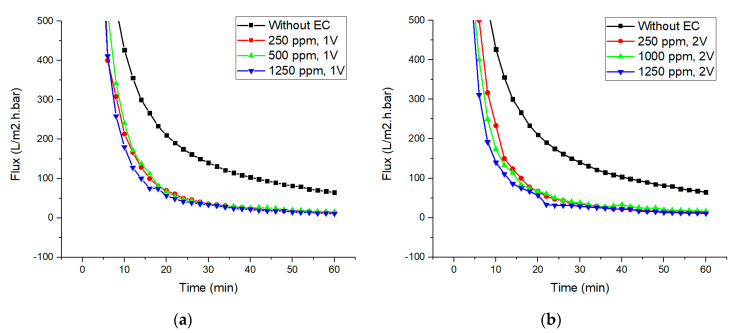
Permeability flux for electrocoagulation-integrated membrane using (**a**) 250 ppm, 500 ppm and 1250 ppm of NaCl at 1V and (**b**) 250 ppm, 1000 ppm and 1250 ppm NaCl at 2 V, constant 10 ppm Celestine blue dye and constant pH 7.

**Table 1 membranes-10-00184-t001:** Properties of the nylon 6,6 nanofiber membrane.

Membrane	Pore Size (µm)	Porosity (%)	Surface Roughness (nm)
Nylon 6,6 nanofiber	0.2	71.3 ± 2.0	56.5 ± 10

**Table 2 membranes-10-00184-t002:** The removal efficiency of low and high voltages during the electrocoagulation of celestine blue [23].

Voltage	Removal Efficiency	Energy Usage (kWh/m^3^)
1 V	43.2%	0.75
2 V	42.0%	1.65
10 V	100.0%	31.88

**Table 3 membranes-10-00184-t003:** Comparison of the proposed method and other methods of treatment of dye in wastewater.

Method	Treatment Mechanism	Influencing Variable	Type of Electrode	pH	Dye Removal	Reference
EC	Physicochemical	Voltage, NaCl concentration	Iron	7	43.2%	[23]
Integrated nanofiber (NF) membrane and EC	Physicochemical and Filtration	Voltage supplied, time	Iron	7	79.4%	This paper
Adsorptive carbon nanofibers	Adsorption	pH, concentration	-	3–11	77.8%	[54]
Integrated bio-electrochemical and anaerobic system	Biological and bio-electrochemical	Reactor configuration and reflux ratio	Graphite	7.27	97.5%	[55]

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
