# Peer review of "Integrated Membrane–Electrocoagulation System for Removal of Celestine Blue Dyes in Wastewater"

_membranes, 2020, doi:10.3390/membranes10080184_

Round 1
Reviewer 1 Report
The authors have prepared an interesting manuscript regarding the treatment of dye contaminated water with a combination of electrocoagulation and membrane filtration. The results they have obtained are interesting and the methodology they have followed seems sound. There are some points in the manuscript that can be improved.
The dye solution used in the experiments should be called “synthetic wastewater” and not wastewater at it was prepared in the lab with standard reagents.
In the description of the experimental procedure a paper filter is mentioned but not included in the diagram of the apparatus. Was the solution pretreated with this filter before being fed to the membrane?
The authors have presented data regarding the flux-drop observed during filtration with and without electrocoagulation, with the combination of the two methods leading to a more severe flux-drop. This can be a significant problem, but maybe the characteristics of the flux-drop are different. If fouling is caused by larger flocs and is mostly on the surface of the membrane, it may be more easily reversible. Have the authors examined the reversibility of flux-drop through washing of the membrane?
Finally, the manuscript has several grammatical errors and the overall English should be improved.
Reviewer 2 Report
The manuscript evaluates an EC-membrane integrated system for Dye removal. Although the concept is interesting and worth publication, the novelty and need for the study are not clear. The manuscript may be considered for publication after addressing the following comments:
- The language of the manuscript could use some improvements. There are a few grammatical errors that should be rectified. The use of “our” in describing previous work is not recommended.
- The authors should clearly indicate the novelty of the study. The integration of EC with membrane separation has been reported before. What is new in this study?
- The authors indicated 79.4% removal of the dye when using both EC and membrane. Is there a synergy when combining the two methods or just like having two steps in series?
- The authors refer to 0.8 g of NaCl. Is this in one liter? It should be reported as a concentration.
- There is no mention of Current Density, which is a key parameter in EC studies. The authors should report the CD values.
- The figures related to the membrane flux (e.g. 3, 4, 5) are of very poor quality and should be improved. No need for Fig. 3.
- The comparison in Table 3 indicates that the Bio-EC is more effective, but the authors indicated it is complex and costly. This is not true, as EC-membrane is more costly than Bio-EC, and membrane fouling is a major issue in membranes.
- There is no statistical analysis or any indication of repeatability of results. The authors used the term “optimized” but they did not use any optimization technique. They should say “best removal” not “optimum removal”.
Reviewer 3 Report
The authors employed the integrated membrane-electrocoagulation system to remove the celestine bule dyes and had achieved some reasonable results. The removal rate was improved due to the additional microfiltration (pore size, 0.2 μm). In integrated membrane-electrocoagulation system, 79.4% dye removal was achieved with only 2 V applied and 1.65 kWh/m3 energy supplied. However, there are still several issues need to be addressed. I would recommend this paper to be published in Membranes after major revision. The detailed comments are listed below:
1. In Abstract part, the authors explained that the no residual Fe concentration was detected. Does all the permeate not contain Fe ion under different operational parameters?
2. Line 82-85, the authors should explain more on the question that why they choose the membrane process after the electrocoagulation method.
3. Line 154, for the integrated system, the outlet of the membrane module for the retentate may need a valve to control the transmembrane pressure (TMP) to be 0.1 bar (10 kPa). Besides, the pressure was quite low for membrane filtration process. For higher TMP, the fouling might be more severe.
4. With respect to Figure 5, the flux obtained a sharp decline in the first 20 min, indicating the membrane fouling was serious. Taking the membrane fouling into consideration, the authors should notice the operational cost in practical application.
5. Following question 3, figure 5 showed that the EC caused more membrane fouling. The relationship between the pore size of membrane and the size of the EC flocs was significant for the control of membrane fouling. If the authors could optimize the flocs to be large and loose, the fouling may be mitigated.
6. Line 274, the word “alas” might be a spelling mistake.
7. The descriptions of the figure captions and the scientific expressions in the manuscript might need to be improved.
8. For Figure 4, the removal rate was detected at various experimental time. Typically, the samples were taken at interval time. However, the curves in Figure 4 was continuous. The authors may describe the detection method detailly.
Round 2
Reviewer 2 Report
The authors have addressed most of my comments. However, the manuscript still has some grammatical errors even in the abstract:
- "The" textile industry.... "The" should be added.
- The textile industry... provide"s". "s" is missing
- The industry also discharge"s". "s" is missing
The whole manuscript should be rechecked before resubmission.